# DNTextSpotter: Arbitrary-Shaped Scene Text Spotting via Improved Denoising Training

Anonymous Paper

## ABSTRACT

More and more end-to-end text spotting methods based on Transformer architecture have demonstrated superior performance. These methods utilize a bipartite graph matching algorithm to perform one-to-one optimal matching between predicted objects and actual objects. However, the instability of bipartite graph matching can lead to inconsistent optimization targets, thereby affecting the training performance of the model. Existing literature applies denoising training to solve the problem of bipartite graph matching instability in object detection tasks. Unfortunately, this denoising training method cannot be directly applied to text spotting tasks, as these tasks need to perform irregular shape detection tasks and more complex text recognition tasks than classification. To relieve the issue, in this paper, we propose a novel denoising training method (DNTextSpotter) for arbitrary-shaped text detection and recognition. Specifically, we decompose the queries of the denoising part into noised positional queries and noised content queries. We use the four Bezier control points of the Bezier center curve to generate the noised positional queries. For the noised content queries, considering that the output of the text in a fixed positional order is not conducive to aligning position with content, we employ a masked character sliding method to initialize content queries to assist the alignment of text content and position. To improve the model's perception of the background, we further utilize an additional loss function for background characters classification in the denoising training part. Although DNTextSpotter is conceptually simple, it outperforms the state-of-the-art methods on four benchmarks (Total-Text, SCUT-CTW1500, ICDAR15, and Inverse-Text), especially yielding an improvement of 11.3% against the best approach in Inverse-Text.

## CCS CONCEPTS

• **Computing methodologies** → **Computer vision problems**;

## KEYWORDS

Scene text spotting, Transformer, Denoising training

## 1 INTRODUCTION

Text Spotter, as an essential foundational technology that encompasses text detection and recognition, plays a critical role in various

Permission to make digital or hard copies of all or part of this work for personal or classroom use is granted without fee provided that copies are not made or distributed for profit or commercial advantage and that copies bear this notice and the full citation on the first page. Copyrights for components of this work owned by others than the author(s) must be honored. Abstracting with credit is permitted. To copy otherwise, or republish, to post on servers or to redistribute to lists, requires prior specific permission and/or a fee. Request permissions from permissions@acm.org.

*ACM MM, 2024, Melbourne, Australia*

© 2024 Copyright held by the owner/author(s). Publication rights licensed to ACM.
ACM ISBN 978-x-xxxx-xxxx-x/YY/MM
https://doi.org/10.1145/nnnnnnn.nnnnnnn

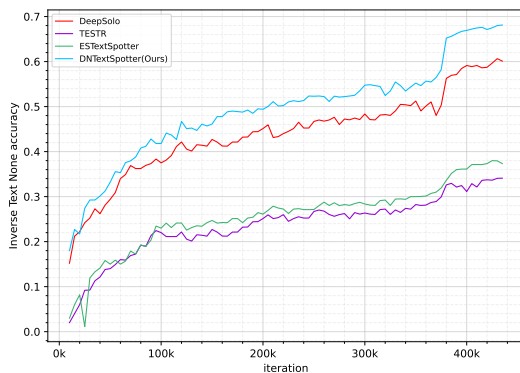

**Figure 1: The convergence curves of DNTextSpotter (Ours), DeepSolo, TESTR, and ESTextSpotter on the Inverse-Text dataset using the ResNet-50 backbone in the 'None' results, where 'None' denotes the F1-measure without lexicon.**

domains [4, 29, 35, 45] such as autonomous driving, security monitoring, and social media analysis. Given that textual elements in wild scenarios are often set against complex backgrounds, presented in diverse font sizes, and subject to distortions, accurate identification of text remains a challenging and dynamic field of research. To address these challenges, traditional CNN-based text spotters [1, 20, 23, 26, 33, 36, 38] divide the text localization task into separate detection and recognition stages, following a detect-then-recognize principle.

Compared to these classical spotting algorithms, building on the Deformable DETR [50], TESTR [48] represents a significant advancement in text spotting with its dual decoder design. This novel approach streamlines the process by removing the necessity for manually designed components and eliminating intermediate steps. TESTR simplifies the complex tasks of detection and recognition by treating them as a unified set prediction problem. It leverages bipartite graph matching to concurrently assign labels for both detection and recognition, achieving a more efficient and integrated workflow. Despite the significant achievements of TESTR, the employment of two decoders significantly increases computational complexity. Moreover, initializing queries with distinct properties for detection and recognition poses challenges for model optimization. Many methods have been proposed to address these issues. For example, TTS [15] attempts to unify the detection and recognition tasks within a single decoder. DeepSolo [44], while using a single decoder, introduces a novel query form with shared parameters, which initializes decoder queries by utilizing a series of explicit coordinates generated from the text line. However, despite further improvements in performance, these methods overlook the instability introduced by the bipartite matching employed in the DETR-like methods. In general object detection tasks,

DN-DETR [16] firstly points out the instability problem of bipartite graph matching when using the DETR architecture and proposes denoising training to solve this problem. Denoising training, in simple terms, initializes noised queries using ground truth with a small amount of noise added, allowing for direct loss calculation with the ground truth after decoding, bypassing the bipartite graph matching algorithm. DINO [46] further proposes a contrastive denoising training method to further enhance the performance of denoising training. Unfortunately, for the tasks of spotting scene text, the challenge is significantly amplified due to the arbitrary shapes of the text to be detected and the need for recognition tasks that are more complex than mere classification. This complexity makes it difficult to directly apply this denoising training method. In fact, ESTextSpotter [12] directly incorporates a DINO-based denoising training method within its model architecture. In this context, as shown by the convergence curves in Fig. 1, this denoising training starts with regular bounding boxes as queries initialization, and the results on inverse-like texts become very poor, indirectly reflecting the negative impact brought by this coarse prior.

In this paper, we propose a novel denoising training method specifically designed for transformer-based text spotters that handle arbitrary shapes. Considering that the task of text spotting aims at the detection and recognition of text in any shape, and using regular boxes to initialize noised queries is coarse, we abandon the traditional approach that relies on 4D anchor boxes and classification labels. Instead, we use Bezier control points and text characters to initialize noised queries. Technically, we feed the noised queries obtained from the ground truth along with randomly initialized learnable queries into the transformer's decoder. We design the noised query using bezier control points of the bezier center curve and text scripts, thereby accomplishing the denoising of both points coordinates and text characters. In addition, considering that outputting the text characters in a fixed positional order is not conducive to aligning position with content, before initializing the text script as noised queries, we use a masked character sliding method to initialize noised content queries to assist in the alignment of content and position of the text instance.

We verify the effectiveness of DNTextSpotter by using multiple public datasets. In the metrics of 'None' results with ResNet-50 backbone, compared with the current state-of-the-art methods, our method achieves 2.0% and 2.1% improved results on the Total Text and CTW1500 datasets respectively, reaching 84.5% and 67.0% respectively. On the newly released benchmark Inverse-Text dataset, our method even exceeds the state-of-the-art results by 11.3%, reaching 75.9%. When switching to the ViTAEv2-S backbone, scores for all metrics are further improved.

In summary, our main contributions can be summarized as follows:

- We introduce a novel denoising training method to design an end-to-end text spotting architecture. Starting from the attribute of arbitrary shapes of scene text, we utilize bezier control points as well as text characters to design this denoising training method.
- Taking into account the negative impact of directly using ground truth text scripts to initialize noised queries, which leads to misalignment between the position of the characters

and the content of these characters, we design a masked character sliding method to preprocess these ground truth text scripts, thereby optimizing the alignment between text position and content.
- Our method achieves state-of-the-art results on multiple benchmarks. Specifically, we conduct a qualitative analysis of several text spotting architectures based on the transformer structure, including analyses of instability results and visualization of results.

## 2 RELATED WORKS

Early literature tends to classify end-to-end text spotting architecture into two-stage methods and one-stage methods. Recently, due to the popularity of transformer-based text spotters, we categorize these methods into CNN-based methods and transformer-based methods. Earlier comprehensive and in-depth surveys on text spotting are available in [4, 29].

### 2.1 CNN-based Text Spotter

The first end-to-end scene text recognition network [17] combines detection and recognition into a single system. This method is limited to recognizing regular-shaped text. Subsequent works [1, 8] improve the connection between the detector and recognizer, considering single characters or text blocks to handle irregular text more flexibly. The Mask TextSpotter series [20, 30] employs segmentation approaches to generate proposals. These methods rely on character-level annotations, significantly increasing the effort required for generating ground truth. Text Perceptron [36] and Boundary [33] utilize Thin-Plate-Spline [2] transformation to rectify features obtained from curved text. The ABCNet series [24, 26] use BezierAlign to address the problem of curved text, requiring the prediction of a small fixed number of points.

While these methods achieve good performance, they require additional RoI-based [9] or TPS-based connectors, and the only shared part between the detector and recognizer is the backbone network's features, neglecting the collaborative nature of detection and recognition. [49] propose ARTS, highlighting the importance of collaborative detection and recognition in the text spotting task. Lastly, all of the aforementioned methods require complex manual operations like Non-Maximum Suppression (NMS).

### 2.2 Transformer-based Text Spotter

With the impressive success of Transformers in visual tasks [6, 7, 14, 18, 27, 28, 42, 47], also influenced by the DETR family [3, 16, 22, 31, 46, 50], more recent works explore Transformer-based structures for the Text Spotting. TESTR [48] employs dual decoders for detection and recognition tasks, sharing the backbone and Transformer encoder features. TTS [15] utilizes an encoder and a decoder with multiple prediction heads for performing multi-tasks. DeepSolo [44] employs an explicit points method to model decoder queries.

Although these methods achieve promising results, they still exhibit certain limitations. The random initialized queries used in TESTR [48] and TTS [15] still lack clarity and fail to efficiently represent queries encompassing both positional and semantic aspects. Utilizing the encoder's output features, DeepSolo [44] generates

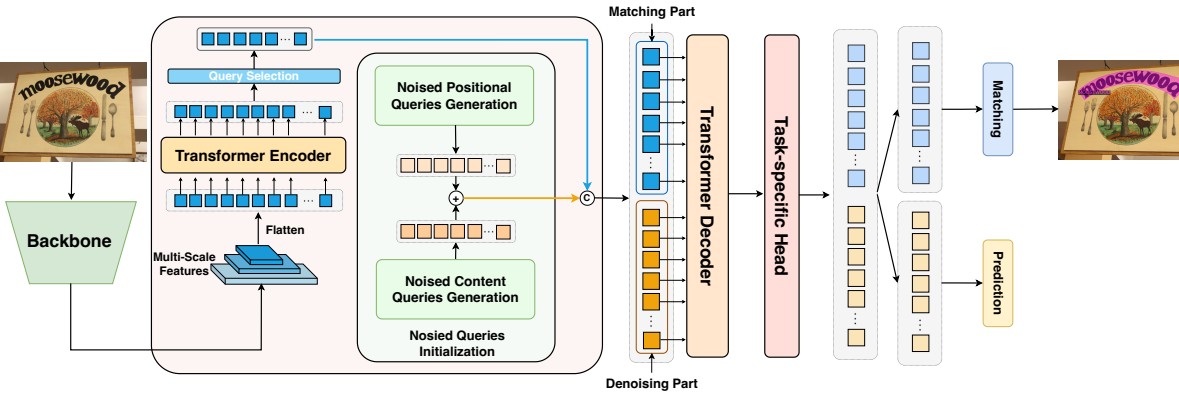

Figure 2: The overall framework of DNTextSpotter. The model utilizes a backbone and an encoder to extract multi-scale features. The queries of the decoder can be divided into two parts: a matching part and a denoising part. The queries in the matching part are randomly initialized queries. The noised queries of denoising part can be found in Fig .3 and Fig .4. After decoder and task-specific head, the matching part calculates loss through a bipartite graph matching algorithm, and the denoising part calculates loss directly with the ground truth.

Bezier center curve proposals, which subsequently serve to generate positional queries. It effectively decouples the ambiguously defined queries into positional queries and content queries. Although these methods have achieved certain accomplishments, they, like DETR-like methods, use bipartite graph matching algorithms to obtain one-to-one matching results. The instability of bipartite graph matching has been proven to have a negative impact on detection tasks by DN-DETR [16] and DINO [46]. While ESTextSpotter [12] attempted to propose a method for denoising training, this method uses boxes as a positional prior for point prediction, failing to consider the irregular attributes of text instances and the characteristics of text scripts. Therefore, we design a denoising training approach based on Bezier control points and characters.

## 3 METHODOLOGY

### 3.1 Preliminaries

**Denoising Training.** The denoising training method was first proposed by DN-DETR to address the slow convergence issue of DETR. This method constructs an additional auxiliary task in the decoder section without bipartite matching, which can be used to accelerate the convergence of DETR-like methods. Technically, it additionally feeds noised ground-truth boxes and labels into the transformer decoder to reconstruct these ground-truths, and this part is updated through an additional auxiliary DN loss. DINO improved the denoising training method and further proposed a contrastive denoising training method, which adds negative queries in addition to the original noised queries to predict the background. In our method, we further extend the denoising training approach based on this foundation.

**Bezier Center Curve.** ABCNet[24] was the first to use Bezier curves to flexibly adapt to any shape of scene text with a small number of fixed points. Subsequently, DeepSolo introduced the Bezier Center Curve, which initializes transformer decoder queries by uniformly sampling a fixed number of points on the curve. This Bezier Center Curve is obtained by calculating the average of the four Bezier control points on the top and bottom edges of each text

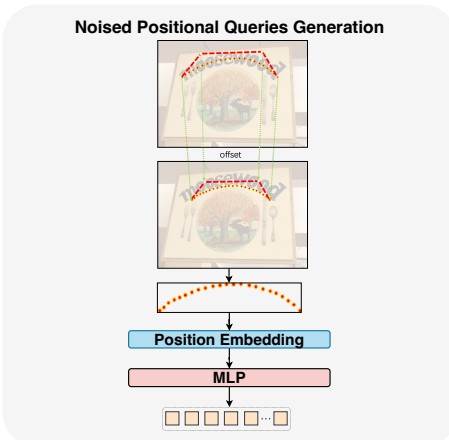

Figure 3: We generate noised positional queries using four Bezier control points from the ground truth, which includes uniformly sampling points along the Bezier curve, position embedding, and a two-layer MLP.

instance. In our method, we consider utilizing Bezier center curves in denoising training.

### 3.2 Query Initialization

The ambiguous meaning of decoder queries in DETR is often interpreted in existing literature[22, 46] as a combination of positional queries and content queries. We also utilize this modeling approach and follow DeepSolo, using the Bezier Center Curve to initialize positional queries and then combine them with learnable content queries. In the denoising part, we represent noised queries as two parts: noised positional queries and noised content queries.

**Noised Positional Queries.** As shown in Fig. 3, for any text instance, the four Bezier control points of the Bezier center curve. Assuming that a picture contains $N$ text instances, we can represent the set of all instances as: $\mathcal{S} = \{S_j | j = 1, 2, \ldots, N\}$, $S_j$ is a set of the four Bezier control points of the Bezier center curve of a text

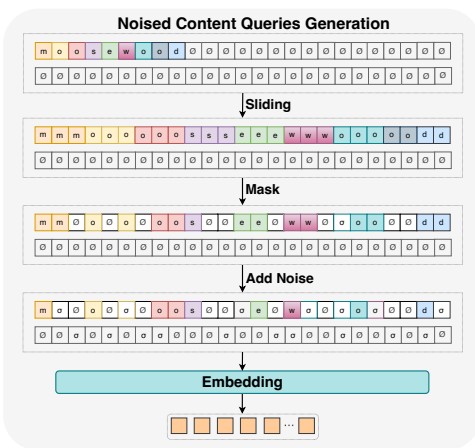

**Figure 4: This image shows the process of generating noised content queries from the ground-truth texts. "∅" indicates the characters will be masked, while "σ" denotes flipping the characters into any character.**

instance. Each instance can be represented as:

$$\mathcal{S}_j = \{(x_{ij}, y_{ij})|i = 0, 1, 2, 3\}, \tag{1}$$

where each point is composed of a pair of coordinates $(x_{ij}, y_{ij})$, and the index $i$ ranges from 0 to 3, indexing the four Bezier control points. Random noise is added to these points to obtain:

$$\mathcal{S}'_j = \{(x_{ij} + \Delta x_{ij}, y_{ij} + \Delta y_{ij})|i = 0, 1, 2, 3\}, \tag{2}$$

where $\Delta x_{ij}$ and $\Delta y_{ij}$ are obtained by calculating the distance between the four Bezier control points of the center curve and the four Bezier control points on the top side and are denoted as $D_{ij}^x$ and $D_{ij}^y$. We use elements $\alpha_{ij}$ and $\beta_{ij}$ from sets that satisfy a $0-1$ uniform distribution to control the noise ratio for the $x$ and $y$ coordinates respectively. Therefore, we can represent the offset of the coordinates as follows:

$$\Delta x_{ij} = \begin{cases} (-1)^m \alpha_{ij} D_{ij}^x & \text{if positive;} \\ (-1)^m (\alpha_{ij} + 1) D_{ij}^x & \text{otherwise.} \end{cases} \tag{3}$$

$$\Delta y_{ij} = \begin{cases} (-1)^m \beta_{ij} D_{ij}^y & \text{if positive;} \\ (-1)^m (\beta_{ij} + 1) D_{ij}^y & \text{otherwise.} \end{cases} \tag{4}$$

where $m$ and $n$ are used to control the direction of the offset, with "positive" indicating the coordinates of the positive part.

After adding noise to the Bezier control points, we obtain a new set of control points $\mathcal{S}'$, Using these noisy Bezier control points, we uniformly sample $T$ point coordinates from the resulting Bezier curves. These point coordinates form tensor Points with shape $(N, T, 2)$. Finally, these coordinates are processed through positional encoding $(PE)$ and two layers of MLP to obtain Noised Positional Queries $(Q_N)$ with a shape of $(N, T, 256)$. We represent $Q_N$ as follows:

$$Q_N = MLP(PE(Points)). \tag{5}$$

Section 4.4 analyzes 'Why add noise to the Bezier control points?'. **Noised Content Queries.** We designed Mask Character Sliding (MCS), as shown in Fig. 4, to initialize the noised content queries.

For the maximum recognition length $T$, which is equal to the $T$ mentioned above, we perform a sliding operation on the valid characters in the positive part. Specifically, we first determine the number of valid characters $t$, which refers to the number of characters in the input sequence that actually have meaning. Then, we calculate the number of times each valid character should be cloned by performing a division operation $T//t$, in order to evenly distribute the total length $T$ of the sequence to each valid character. Additionally, since the division of $T$ by $t$ may not be exact, there will be a remainder $k = T - T//t$, indicating that there are $k$ additional spaces that need to be allocated. To fairly distribute these extra spaces, we assign one additional clone to each of the first $k$ valid characters in the sequence, ensuring that the allocation for each character is as even as possible. Since this operation visually resembles a character sliding operation, we name it Character Sliding. After sliding the characters, we use a mask operation to control the number of consecutive characters, flipping a portion of the consecutive characters into a background label with a certain probability. After processing the positive part, we add noise to the characters in both the positive part and the negative part, causing these characters to flip to other characters with a probability of $\lambda$. These characters are then transformed into noised content queries after being embedded, with all the characters initialized in the negative part being backgrounds.

In addition, we use a dynamic group $g$ to fully utilize the performance of denoising training. Considering the computational cost, we set the maximum number of text instances $N$ per image to 100. When $B$ exceeds 100, we simply use the slicing method to take the first 100 instances. The division of $g$ is as follows:

$$g = (5, \left\lfloor \frac{100}{n} \right\rfloor). \tag{6}$$

## 3.3 Single Attention Mask

To ensure that during the decoder self-attention calculation, the information in the denoising part contains ground-truth information, we need to ensure that the information in the matching part cannot see the information in the denoising part. In addition, each group should not be able to see each other. Considering that there are two parts during self-attention calculation, namely intra-relation self-attention which calculates the attention relationship between characters, and inter-relation self-attention which calculates the attention relationship between text instances. We consider whether two attention masks are needed to prevent information leakage. However, in fact, for intra-relation self-attention which calculates the attention relationship between characters, we do not need to use an attention mask, because, for any text instance (including the denoising part and the matching part), the attention calculation is within the text instance and does not interact with other text instances. So we only need to consider the design of inter-relation self-attention, we call this attention mask that only needs to be used once as Single Attention Mask, and we devise the attention mask $\mathbf{A} = \left[\mathbf{a}_{ij}\right]_{(g+2n)\times(g+2n)}$ as follows:

$$a_{ij} = \begin{cases} 1, & \text{if } j < g \times 2n \text{ and } \lfloor \frac{i}{2n} \rfloor \neq \lfloor \frac{j}{2n} \rfloor; \\ 1, & \text{if } j < g \times 2n \text{ and } i \geq g \times 2n; \\ 0, & \text{otherwise.} \end{cases} \tag{7}$$

where $g$ and $n$ represent the number of groups and the number of text instances per image, respectively. $a_{ij} = 1$ means the $i$-th query is blind to the $j$-th query; if it's 0, they can see each other.

## 3.4 Training Losses

Compared to the matching part, the denoising part uses a slightly modified loss function and an additional background computation loss function (namely, the background part is calculated twice, which we briefly refer to as BCT). The rest of the loss functions are consistent with DeepSolo [44], including the Hungarian matching algorithm, focal loss, CTC loss [10], and L1 loss. The details of the matching part are elaborated in the appendix.

**Overall Loss**. We use focal loss [21] to calculate the classification of text instances. In each set of denoising queries, the positive part represents positive samples, and the negative part represents negative samples, with the focal loss being used to compute the background loss for the first time. Therefore, for the $\tau^{th}$ query in the positive part of the denoising queries, the calculation of the focal loss for text instance classification is as follows:

$$\mathcal{L}_{\text{cls}}^{(\tau)} = -\mathbb{1}_{\{\tau \in \text{Im}(\varphi)\}}\alpha(1 - \hat{b}^{(\tau)})^{\gamma}\log(\hat{b}^{(\tau)})$$
$$- \mathbb{1}_{\{\tau \notin \text{Im}(\varphi)\}}(1 - \alpha)(\hat{b}^{(\tau)})^{\gamma}\log(1 - \hat{b}^{(\tau)}), \quad (8)$$

where $\mathbb{1}$ represents the indicator function, and $\text{Im}(\varphi)$ denotes the image of the mapping $\varphi$. Concerning character classification, for the $\tau$-th denoised query in the positive part, we employ the CTC loss:

$$\mathcal{L}_{\text{text,pos}}^{(\tau)} = \mathbb{1}_{\{\tau \in \text{Im}(\varphi)\}}\text{CTC}(t^{(\varphi^{-1}(\tau))}, \hat{t}^{(\tau)}). \quad (9)$$

The cross entropy loss for the $\kappa$-th denoised query in the negative part during the second background calculation:

$$\mathcal{L}_{\text{text,neg}}^{(\kappa)} = \mathbb{1}_{\{\kappa \in \text{Im}(\varphi)\}}\text{CE}(t^{(\varphi^{-1}(\kappa))}, \hat{t}^{(\kappa)}). \quad (10)$$

Additionally, for the coordinate points of the center curve and boundaries in the positive part of the $\tau$-th denoised query, we employ the L1 loss for the computation:

$$\mathcal{L}_{\text{coord}}^{(\tau)} = \mathbb{1}_{\{\tau \in \text{Im}(\varphi)\}}\sum_{n=0}^{N-1}\left\|p_n^{(\varphi^{-1}(\tau))} - \hat{p}_n^{(\tau)}\right\|, \quad (11)$$

$$\mathcal{L}_{\text{bd}}^{(\tau)} = \mathbb{1}_{\{\tau \in \text{Im}(\varphi)\}}\sum_{n=0}^{N-1}\left(\left\|top_n^{(\varphi^{-1}(\tau))} - t\hat{o}p_n^{(\tau)}\right\| + \left\|bot_n^{(\varphi^{-1}(\tau))} - b\hat{o}t_n^{(\tau)}\right\|\right), \quad (12)$$

where *top* refers to the top curves of the boundaries, and *bot* refers to the bottom curves of the boundaries. The negative part does not participate in the calculation of the part.

The loss function for the denoised queries consists of four aforementioned losses in the positive part and two aforementioned losses in negative part:

$$\mathcal{L}_{\text{pos}} = \sum_{\tau}\left(\lambda_{\text{cls}}\mathcal{L}_{\text{cls}}^{(\tau)} + \lambda_{\text{text,pos}}\mathcal{L}_{\text{text,pos}}^{(\tau)} \right.$$
$$\left. + \lambda_{\text{coord}}\mathcal{L}_{\text{coord}}^{(\tau)} + \lambda_{\text{bd}}\mathcal{L}_{\text{bd}}^{(\tau)}\right), \quad (13)$$

$$\mathcal{L}_{\text{neg}} = \sum_{\kappa}\left(\lambda_{\text{cls}}\mathcal{L}_{\text{cls}}^{(\kappa)} + \lambda_{\text{text,neg}}\mathcal{L}_{\text{text,neg}}^{(\kappa)}\right), \quad (14)$$

where $\lambda_{\text{cls}}$, $\lambda_{\text{text}}$, $\lambda_{\text{coord}}$, $\lambda_{\text{bd}}$ are hyper-parameters to balance different tasks. The final loss function of the denoising part is:

$$\mathcal{L}_{\text{dn}} = \mathcal{L}_{\text{pos}} + \mathcal{L}_{\text{neg}}. \quad (15)$$

## 4 EXPERIMENT

We conduct comparisons with known Transformer-based approaches on various datasets, including Total-Text, SCUT-CTW1500, IC-DAR15, and InverseText which contain multi-directional scene text and arbitrary-shaped text instances. In addition, We choose publicly available datasets Synth150K, MLT17, IC13, and TextOCR as additional pre-training datasets.

## 4.1 Public Datasets

**Total-Text** is a widely used comprehensive scene text dataset introduced by [25], specifically designed for arbitrary text detection. It comprises 1255 training images and 300 testing images, containing horizontal, multi-directional, and arbitrary-shaped text instances.
**SCUT-CTW1500** is another significant dataset for arbitrary-shaped text, published by [5]. It comprises 1500 images, consisting of 1000 training images and 500 testing images.
**ICDAR2015 Incidental Text (ICDAR15)** [13] includes 1000 training images and 500 testing images with quadrilateral text. It contains multi-directional text instances annotated with word-level quadrilateral annotations.
**InverseText** was manually annotated by [43] and includes 500 test images. Unlike the previous datasets, this dataset contains 40% inverse-like text instances, specifically designed to address the lack of inverse-like texts in existing test datasets.

## 4.2 Implementation Details

All settings are based on the ResNet-50 backbone. We employ 6 layers of encoder and 6 layers of decoder, with a hidden dimension of 256. For character classification, we predict 37 classes on the Total Text, ICDAR15 , and InverseText datasets, and 96 classes on the CTW1500 dataset. During training, we set the noise hyperparameters as follows: The probability $\lambda$ of characters being flipped to other characters is set to 0.4. The learning rate scheduler utilizes an initial learning rate of $2e^{-5}$ for the backbone and $2e^{-4}$ for other parts. We train the DNTextSpotter for a total of $435k$ steps, and the learning rate is reduced by a factor of 0.1 at $375k$ steps. We use AdamW as the optimizer and train our network with a batch size of 8. The denoising part loss weights $\lambda_{\text{cls}}$, $\lambda_{\text{coord}}$, $\lambda_{\text{bd}}$, $\lambda_{\text{text,pos}}$, and $\lambda_{\text{text,neg}}$ are set to 1.0, 1.0, 0.5, 0.5, and 0.5. The focal loss parameters $\alpha$ and $\gamma$ are set to 0.25 and 2.0, respectively.

## 4.3 Comparison with State-of-the-Art Methods

**For Arbitrarily-Shaped Scene Text Spotting:** As previously indicated, the Total-Text and SCUT-CTW1500 datasets are specifically designed to emphasize text instances characterized by arbitrary shapes. In comparison to other methods on the Total-Text dataset (shown in Table 1), in the detection task, our approach is close to the current state-of-the-art method, ESTextSpotter, achieving 89.2%. While its detection performance is slightly lower, it has a significant advantage in recognition performance. Without a lexicon ('None' results), outperforming ESTextSpotter by 3.7%, reaching

Table 1: Performances on Total-Text and CTW1500 with different backbone. "None" denotes lexicon-free. "Full" denotes the inclusion of all words present in the test dataset. The top three scores are shown in bold red, blue, and green fonts.

| Method | Backbone | Total Text | | | | | CTW1500 | | | | | FPS |
|---|---|---|---|---|---|---|---|---|---|---|---|---|
| | | Detection | | | E2E | | Detection | | | E2E | | |
| | | P | R | F | None | Full | P | R | F | None | Full | |
| TextDragon [8] | VGG16 | 85.6 | 75.7 | 80.3 | 48.8 | 74.8 | 84.5 | 82.8 | 83.6 | 39.7 | 72.4 | − |
| SRSTS [40] | ResNet-50 | 92.0 | 83.0 | 87.2 | 78.8 | 86.3 | − | − | − | − | − | 18.7 |
| CharNet [41] | ResNet-50-Hourglass57 | 88.6 | 81.8 | 84.6 | 63.6 | − | − | − | − | − | − | 1.2 |
| TextPerceptron [33] | ResNet-50-FPN | 88.8 | 81.8 | 85.2 | 69.7 | 78.3 | − | − | − | − | 57.0 | − |
| Boundary [36] | ResNet-50-FPN | 88.9 | 85.0 | 87.0 | 65.0 | 76.1 | − | − | − | − | 46.1 | 73.0 |
| PGNet [37] | ResNet-50-FPN | 85.5 | 86.8 | 86.8 | 63.1 | − | − | − | − | − | − | 35.5 |
| ABCNet v2 [26] | ResNet-50-FPN | 90.2 | 84.1 | 87.0 | 70.4 | 78.1 | 85.6 | 83.8 | 84.7 | 57.5 | 77.2 | 10.0 |
| TPSNet [39] | ResNet-50-FPN | 90.2 | 86.8 | 88.5 | 78.5 | 84.1 | 88.7 | 86.3 | 87.5 | 60.5 | 80.1 | 14.3 |
| GLASS [34] | ResNet-50-FPN | 90.8 | 85.5 | 88.1 | 79.9 | 86.2 | − | − | − | − | − | 3.0 |
| SwinTextSpotter [11] | Swin-T-FPN | − | − | 88.0 | 74.3 | 84.1 | − | − | 88.0 | 51.8 | 77.0 | 2.9 |
| UNITS[14] | Swin-B | − | − | 89.8 | 78.7 | 86.0 | − | − | − | − | − | |
| TESTR [48] | ResNet-50 | 93.4 | 81.4 | 86.9 | 73.3 | 83.9 | 92.0 | 82.6 | 87.1 | 56.0 | 81.5 | 5.5 |
| TTS [15] | ResNet-50 | − | − | − | 78.2 | 86.3 | − | − | − | − | − | − |
| SPTS [32] | ResNet-50 | − | − | − | 74.2 | 82.4 | − | − | − | 63.6 | 83.8 | 0.4 |
| ESTextSpotter [12] | ResNet-50 | 92.0 | 88.1 | 90.0 | 80.8 | 87.1 | 91.5 | 88.6 | 90.0 | 64.9 | 83.9 | 4.3 |
| DeepSolo [44] | ResNet-50 | 93.2 | 84.6 | 88.7 | 82.5 | 88.7 | 92.5 | 86.3 | 89.3 | 64.2 | 81.4 | 17.0 |
| DeepSolo [44] | VITAEv2-S | 92.9 | 87.4 | 90.0 | 83.6 | 89.6 | − | − | − | − | − | 10.0 |
| DNTextSpotter(Ours) | ResNet-50 | 91.5 | 87.0 | 89.2 | 84.5 | 89.8 | 93.5 | 87.1 | 90.2 | 67.0 | 84.2 | 17.0 |
| DNTextSpotter(Ours) | VITAEv2-S | 92.9 | 88.6 | 90.7 | 85.0 | 90.5 | 94.2 | 88.9 | 91.5 | 69.2 | 85.9 | 10.0 |

Table 2: Performance on Inverse-Text. E2E: the end-to-end spotting results. The top three scores are shown in bold red, blue, and green fonts.

| Method | E2E | |
|---|---|---|
| | None | Full |
| MaskTextSpotter v2 [19](ResNet-50-FPN) | 39.0 | 43.5 |
| ABCNet [24](ResNet-50-FPN) | 22.2 | 34.3 |
| ABCNet v2 [26](ResNet-50-FPN) | 34.5 | 47.4 |
| TESTR [48](ResNet-50) | 34.2 | 41.6 |
| SwinTextSpotter [11](Swin-T-FPN) | 55.4 | 67.9 |
| SPTS [32](ResNet-50) | 38.3 | 46.2 |
| ESTextSpotter [12](ResNet-50) | 51.2 | 55.1 |
| DeepSolo[44] (ResNet-50) | 64.6 | 71.2 |
| DeepSolo[44] (ViTAEv2-S) | 68.8 | 75.8 |
| DNTextSpotter(ResNet-50) | 75.9 | 81.6 |
| DNTextSpotter(ViTAEv2-S) | 78.1 | 83.8 |

Table 3: Performance on ICDAR15. E2E: the end-to-end recognition results. "S," "W", and "G" correspond to Strong, Weak, and Generic lexicon, respectively. The top three scores are shown in bold red, blue, and green fonts.

| Method | E2E | | |
|---|---|---|---|
| | S | W | G |
| ABCNet v2 [26](ResNet-50-FPN) | 82.7 | 78.5 | 73.0 |
| SwinTextSpotter [11](Swin-T-FPN) | 83.9 | 77.3 | 70.5 |
| TESTR [48](ResNet-50) | 85.2 | 79.4 | 73.6 |
| SPTS [32](ResNet-50) | 77.5 | 70.2 | 65.8 |
| ESTextSpotter [12](ResNet-50) | 87.5 | 83.0 | 78.1 |
| DeepSolo [44](ResNet-50) | 88.0 | 83.5 | 79.1 |
| DeepSolo [44](ViTAEv2-S) | 88.1 | 83.9 | 79.5 |
| DNTextSpotter(ResNet-50) | 88.7 | 84.3 | 79.9 |
| DNTextSpotter(ViTAEv2-S) | 89.4 | 85.2 | 80.6 |

84.5%, and with a lexicon ('Full' results), exceeding it by 2.7%, achieving 89.8%. Compared to the state-of-the-art method in recognition performance, DeepSolo, is higher by 2.0% and 1.1%, respectively. On the CTW1500 dataset, both detection and recognition performances have reached the current state-of-the-art. The detection F1 score reached 90.2%, surpassing the state-of-the-art method ESTextSpotter by 0.2%. Without a lexicon ('None' results), it exceeds ESTextSpotter by 2.1%, and with a lexicon ('Full' results), it surpasses by 0.3%, reaching 84.2%. Compared to our baseline model, DeepSolo, there is a significant improvement in performance for detection and recognition, with increases in F1, 'None', and 'Full' by 0.9%, 2.8%, and 2.8%, respectively. We use only ResNet-50 as the backbone, and the experimental results on various datasets reach state-of-the-art. When we switch the backbone to ViTAEv2-S, our performance greatly exceeds that of DeepSolo, which also uses ViTAEv2-S.

**For Arbitrarily-Oriented Scene Text Spotting:** In the case of the multi-oriented benchmark ICDAR15, DNTextSpotter exhibits excellent performance when compared to other Transformer-based methods. As shown in Table 3, DNTextSpotter achieves results of

**Table 4: Comparison of DNTextSpotter results (black) and DeepSolo results (blue) at different Training Steps. The learning rate decays by a factor of 0.1 at 375K steps.**

| Training Steps | InverseText | | TotalText | |
|---|---|---|---|---|
| | None | Full | None | Full |
| 25K | 27.9 (25.2) | 38.2 (34.2) | 52.5 (52.5) | 68.9 (65.3) |
| 50K | 32.1 (29.5) | 35.5 (35.3) | 65.0 (60.0) | 74.2 (70.3) |
| 100K | 40.9 (39.2) | 49.5 (47.3) | 67.9 (64.9) | 79.9 (77.8) |
| 200K | 48.8 (46.5) | 55.8 (52.3) | 73.1 (71.9) | 82.9 (82.2) |
| 300K | 54.5 (48.6) | 61.8 (54.4) | 74.0 (71.5) | 84.1 (82.0) |
| 375K | 58.7 (50.6) | 69.4 (55.1) | 74.9 (73.9) | 83.4 (81.9) |
| 435K | 67.9 (59.1) | 74.4 (65.8) | 79.2 (76.6) | 85.1 (84.0) |

88.7%, 84.3%, and 79.9% on the settings of "S", "W", and "G", respectively. These results surpass the current state-of-the-art method, DeepSolo, by 0.6% in "S", 0.4% in "W", and 0.4% in "G", respectively.

**For Inverse-like Scene Text Spotting:** Besides ESTextSpotter, whose weights were measured using the publicly available weights from the paper, all other results were taken from the DeepSolo report. In the latest Inverse-Text dataset, our method achieves significant success. Compared to the current state-of-the-art method, DeepSolo, our results without a lexicon ("None") surpassed it by 11.3%, reaching 75.9%, and with a lexicon ("Full"), we exceeded it by 10.4%, achieving 81.6%. We analyze why our method performed exceptionally well on this dataset. We believe this is due to the fact that datasets for Inverse-like texts present a more complex challenge than those for arbitrarily shaped texts. Unlike conventional texts that follow a left-to-right order, these texts are ordered from right to left and are flipped downward, making it challenging for the original model to learn this pattern. Denoising training, as an auxiliary task, with its relatively simpler nature, can more easily help the model learn these uncommon or complex forms of text.

**Table 5: Ablation comparison of different noise scales $\lambda$ in the experiment, where $\lambda$ represents the probability of a character flipping to another character.**

| $\lambda$ | Detection | | | E2E | |
|---|---|---|---|---|---|
| | P | R | F1 | None | Full |
| 0.8 | 91.8 | 85.1 | 88.2 | 82.9 | 88.8 |
| 0.6 | 92.2 | 85.6 | 88.7 | 83.4 | 89.2 |
| 0.4 | 91.5 | **87.0** | **89.2** | **84.5** | **89.8** |
| 0.2 | 92.9 | 84.4 | 88.4 | 84.0 | 89.6 |
| 0.0 | 91.2 | 84.8 | 87.8 | 82.7 | 88.6 |

## 4.4 Ablation Studies

Ablation experiments were conducted on the Total-Text dataset. Table 4 further demonstrates the convergence effects of the improved denoising training. Table 6 shows the impact of noise scale and mask probability. Table 7 shows the effects of adding noise to Bezier control points, using masked character sliding, and calculating an additional background loss.

**Table 6: Adjustment of mask probability, where 'mask' refers to the probability of converting repeated characters into the background.**

| Mask Probability | 0% | 25% | 50% | 75% |
|---|---|---|---|---|
| F1 (%) | 88.7 | 88.6 | **89.2** | 88.9 |
| None (%) | 83.1 | 83.7 | **84.5** | 82.9 |
| Full (%) | 88.6 | 88.9 | **89.8** | 88.4 |

**Table 7: Ablation comparison of the proposed components. "DN" denotes whether denoising training is employed. "BCP" refers to the addition of noise to Bézier control points rather than directly to the sampling points on the Bézier curve's centerline. "MCS" denotes the use of masked character sliding, while "BCT" indicates the use of an additional background loss calculation, i.e., background calculation twice.**

| DN | BCP | MCS | BCT | Detection | | | E2E | |
|---|---|---|---|---|---|---|---|---|
| | | | | P | R | F1 | None | Full |
| ✗ | ✗ | ✗ | ✗ | 93.2 | 84.6 | 88.7 | 82.5 | 88.7 |
| ✓ | ✗ | ✗ | ✗ | 91.5 | 85.1 | 88.1 | 82.1 | 88.0 |
| ✓ | ✓ | ✗ | ✗ | 92.3 | 85.9 | 88.9 | 82.5 | 88.4 |
| ✓ | ✓ | ✓ | ✗ | **93.2** | 85.5 | 89.0 | 84.1 | 89.2 |
| ✓ | ✓ | ✓ | ✓ | 91.5 | **87.0** | **89.2** | **84.5** | **89.8** |

**(1) Effect of BCP:** We investigate the addition of noise on the Bezier control points (BCP) rather than directly on the sampling points of the Bezier center curve. Using BCP leads to a noticeable improvement in 'F1' results, by approximately 0.8%. We think BCP can contribute positional prior information to the noised positional queries, which is not achieved by directly adding noise to the sampling points on the centerline. Conversely, if noise is directly added to these sampling points, the denoising training segment would miss out on the benefits of the smooth Bezier curve's positional priors, thereby negatively impacting the training outcome.

**(2) Effect of MCS:** Generating queries directly without masked character sliding (MCS) leads to a significant decrease in performance because it forces the model to learn a fixed positional output order. Introducing the sliding operation means that the model no longer learns targets based on a fixed position, which promotes a one-to-one alignment between character and position. The experimental results also confirm the effectiveness of MCS, showing a 1.6% increase in 'None' results.

**(3) Effect of BCT:** Employing additional background loss calculation techniques (BCT) leads to improvements in both F1 scores (+0.2%) and 'None' results (+0.4%). BCT is actually a reuse of the negative part. Originally, the focal loss was used solely to calculate the loss for binary classification between the foreground and background. Now, an additional cross-entropy loss requires that each character in the negative part undergo multi-class classification.

**(4) Effect of Noise Scale and Mask Probability:** The ablation experiments presented in Table 5 and Table 6 indicate that choosing an appropriate noise ratio is crucial, as both excessive and insufficient noise can impact the experimental outcomes. We conduct ablation

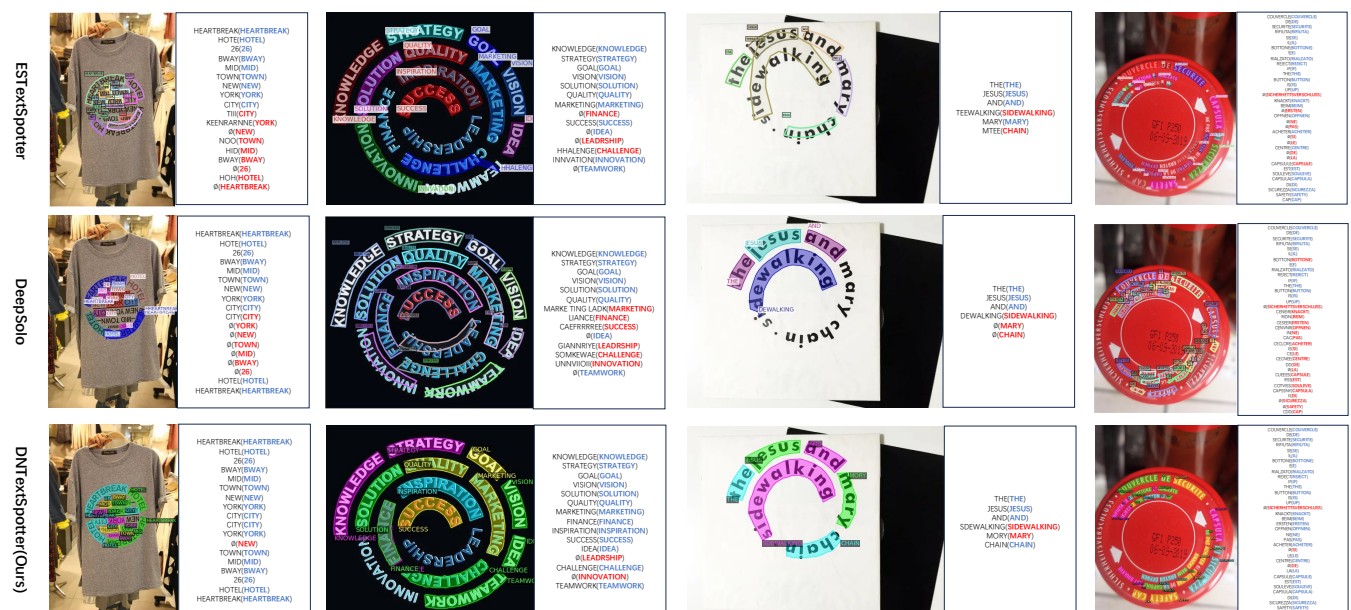

**Figure 5: Several instance examples: rows display ESTextSpotter, DeepSolo, and DNTextSpotter (Ours) visualizations, respectively. In the recognition results, blue within parentheses represents correct recognition, while red denotes incorrect ones; outside the parentheses, ∅ signifies no detection or no recognition took place. Additional visual analysis is provided in the appendix.**

experiments on the hyperparameter $\lambda$ for randomly flipping characters, with the experimental results showing the performance when $\lambda$ is set between 0.0 and 0.8. Furthermore, we control the noise scale $\lambda$ at 0.4 to conduct ablation experiments on the mask probability. The experimental results show that either excessive or insufficient noise can affect the model's performance.

### 4.5 Qualitative Analysis

**Instability Measurement:** We utilize the analysis method for quantifying the instability(*IS*) of bipartite graph matching proposed by DN-DETR[16]. The calculation formula can be found in the appendix. For a training image in the TotalText dataset, we calculate the indices of $N$ proposals every 10k iterations, with every 10k iterations considered as one group. By comparing the differences in indices between the ith group and the $(i + 1)$th group, we obtain the results for *IS*. We visualize the *IS* results as shown in Fig. 6. Additionally, the training set of the Total Text contains a total of 1255 training images, with an average of 7.04 text instances per image, so the largest possible *IS* is $7.04 \times 2 = 14.08$.

**Visualization Comparisons:** Fig. 5 illustrates the experimental results on the InverseText dataset. ESTextSpotter significantly struggles with the recognition of inverse-like text. Even for some texts, distortions and deformations occur during detection. Although DeepSolo has greatly improved the recognition of these inverse-like texts, it faces challenges in recognizing all the more dense texts. As for DNTextSpotter, we achieve good performance on most of the inverse-like texts, indicating that denoising training has an increasingly positive effect on more complex tasks. More detailed visualization results can be found in the appendix.

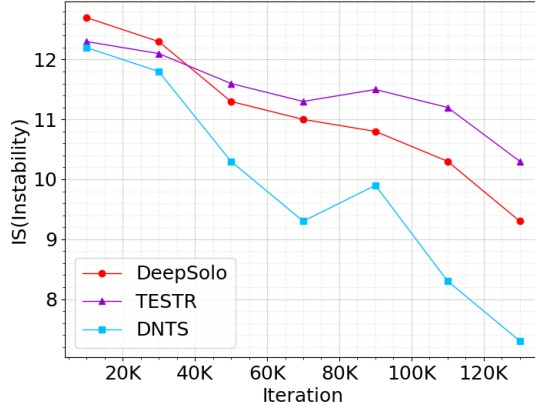

**Figure 6: For the IS of TESTR, DeepSolo, and DNTextSpotter, we trained for 120k steps under the same settings, calculating the IS at every consecutive 10k step interval.**

## 5 CONCLUSION

In this paper, we design a novel denoising training method from the perspective of the attributes of scene text. Our research has found that devising a denoising training method that aligns the positions and contents of characters is very effective for performance improvement. In the future, designing denoising training methods that align more closely with task characteristics to improve model performance is a promising research avenue. We hope our denoising training approach for text spotters provides valuable insights to other researchers.

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
