# OpenReview forum: "DNTextSpotter: Arbitrary-Shaped Scene Text Spotting via Improved Denoising Training"
_acmmm.org/ACMMM/2024/Conference — MM2024 Poster_

### Official Review · Reviewer_1Kip · 2024-05-17

**Rating:** 4
**Confidence:** 4

**Summary:**

This paper proposed a denoising training method for arbitrary-shaped text detection and recognition. It decomposed the queries of the denoising part into noised positional queries and noised contexnt queries. For the noised positional queries, the gour Bezier control points of the Bezier center curve is used. For the noised content queries, a masked character sliding method is adopted to align the text content. The proposed DNTextSPotter achieves SOTA results on standard scene text spotting benchmarks.

**Strengths:**

Innovative Application: The paper's primary contribution lies in the application of a denoising training strategy to the challenging task of spotting arbitrary-shaped text. This is a novel approach that addresses the need for complex positional descriptions and aligned ground truths in text recognition.


Technical Clarity: The motivation behind the proposed method is clearly articulated, and the paper provides a solid theoretical foundation for its approach. This clarity is crucial for readers to understand and potentially build upon the work.


Empirical Validation: The experimental results are compelling, showing that the proposed modules significantly enhance the performance of text spotting systems. Achieving SOTA results on standard benchmarks is a strong validation of the method's effectiveness.

**Limitations:**

Comparison Clarity: The review suggests that including the training data used by the compared methods in Table 1 would enhance the clarity of the comparisons. This information is essential for understanding the context and fairness of the comparisons made.

**Suitability:**

3

---

### Official Review · Reviewer_ZMok · 2024-05-24

**Rating:** 3
**Confidence:** 4

**Summary:**

This paper presents a new denoising training method for scene text spotting. By making a few modifications to the original denoising training, the proposed method can be used in the Transformer-based spotter DeepSolo. Experiments demonstrate the effectiveness of the proposed method. Although promising results are achieved, the proposed approach appears to be limited to working only with DeepSolo. Additionally, the comparison with previous methods may not be fair.

**Strengths:**

This paper is well-written and well-organized. The idea of improving performance through denoising training is interesting. The proposed method achieves state-of-the-art performance on several scene text spotting benchmarks.

**Limitations:**

1. The denoising training proposed in DN-DETR[16] can help improve the performance under different frameworks. However, it seems like the proposed method can only be used on DeepSolo. This limits the applicability of this approach. Additionally, the proposed denoising training seems to simply modify from [16,46] to adapt it to the DeepSolo.

2. Lack of a fair comparison with other methods. For instance, TextOCR was not used as training data in the most of previous methods. The baseline, DeepSolo, also includes the results without the use of TextOCR. It would be better to include these results. Additionally, the higher performance on Inverse-Text may be due to the use of TextOCR.

3. The proposed method uses TextOCR as a training set but lacks results evaluated on TextOCR.

4. The ablation study is insufficient. The authors spend a section discussing the Single Attention Mask, but there is no ablation study about it.

**Suitability:**

2

---

### Official Review · Reviewer_Wk2s · 2024-06-06

**Rating:** 4
**Confidence:** 3

**Summary:**

This paper proposes DNTextSpotter, a modified denoising training method to  solve the problem of bipartite graph matching instability  for transformer-based text spotting. The main motivation comes from that the DETR methods use a bipartite graph matching algorithm during training, but it brings instability, which leads to slow convergence and suboptimal performance. Inspired by the denoising training method proposed by DN-DETR,  DNTextSpotter makes some improvements to the denoising training method, making it suitable for text spotting scenarios.

**Strengths:**

1. DNTextSpotter extends the denoising training approach to the text spotting domain, achieving SOTA performance and accelerating the convergence speed.
2. DNTextSpotter proposes a masked character sliding method to preprocess ground truth text scripts to align text position and content. It is reasonable.
3. The experiments are relatively sufficient. Besides, the authors conduct a qualitative analysis of transformer-based text spotting architectures, which is insightful.

**Limitations:**

1. The Denoising training method is proposed by DNDETR,  and Bezier Curve is widely-used by previous work to model arbitrarily shaped text representation. Although the denoising method doesn't work directly in the text spotting domain, the improvements made in the paper are  somewhat small and the enhancements are limited, as shown in Table 7. I'm a little concerned about whether the novelty is enough to match MM's criteria.

2. Lack of citations to some related works published in ACM MM. e.g. [1], [2], [3]

[1] Textblock: Towards scene text spotting without fine-grained detection
[2] You Can even Annotate Text with Voice: Transcription-only-Supervised Text Spotting
[3] You Only Recognize Once: Towards Fast Video Text Spotting

**Suitability:**

2

---

### Meta-Review · Area_Chair_oyEt · 2024-07-01

**Recommendation:** Accept (Poster)
**Confidence:** 5

**Metareview:**

The authors did a good rebuttal. The reviewers unanimously recommend acceptance. After checking the rebuttal, the review, and the paper, the AC agrees with this assessment.